# Emotional Impact and Perception of Support in Nursing Home Residents during the COVID-19 Lockdown: A Qualitative Study

**DOI:** 10.3390/ijerph192315712

**Published:** 2022-11-25

**Authors:** Almudena Crespo-Martín, Domingo Palacios-Ceña, Elisabet Huertas-Hoyas, Javier Güeita-Rodríguez, Gemma Fernández-Gómez, Jorge Pérez-Corrales

**Affiliations:** 1Occupational Therapy Department, El Conquistador Nursing Home, 10200 Trujillo, Spain; 2Research Group of Humanities and Qualitative Research in Health Science, Department of Physical Therapy, Occupational Therapy, Physical Medicine and Rehabilitation, Universidad Rey Juan Carlos, 28922 Alcorcon, Spain; 3Research Group in Evaluation and Assessment of Capacity, Functionality and Disability (TO+IDI), Department of Physical Therapy, Occupational Therapy, Physical Medicine and Rehabilitation, Universidad Rey Juan Carlos, 28922 Alcorcon, Spain

**Keywords:** COVID-19, nursing homes, occupational therapy, qualitative research

## Abstract

Social isolation measures implemented in nursing homes during the COVID-19 pandemic generated occupational imbalance, discomfort, and mental health impairment in residents. We aimed to analyze the lived experience of elderly nursing home residents during the lockdown and social contact restrictions resulting from the COVID-19 pandemic. Methods: Exploratory qualitative study. Information was collected through in-depth interviews and field notes. An inductive thematic analysis was performed and international recommendations for the development of qualitative studies were followed. Results: Twenty-four participants residing in nursing homes were included. Two main themes were identified: (1) emotional impact of the experience of COVID-19 lockdown (subthemes: experience of contradictory feelings; illness and death; importance of routine; feeling busy; and role of religious beliefs); and (2) support as a therapeutic tool (subthemes: family support; peer support; and professional support). Conclusion: Social restrictions by COVID-19 caused significant changes in residents’ occupations and routines, producing fear, loneliness, and abandonment of desired occupations; however, very important supports were also identified that helped to overcome the lockdown, such as social support, spirituality, and gratitude.

## 1. Introduction

SARS-CoV-2 is an acute respiratory illness, a causative agent of Coronavirus 2019 (COVID-19) [1]. On 11 March 2020, the World Health Organization declared a global pandemic for COVID-19 [1]. The clinical symptoms of this disease vary from mild to severe illness. From January 2020 to December 2021, an estimated 18.2 million people worldwide have died due to the pandemic [2].

Worldwide [3] and in Spain [4], social isolation measures were applied. The measures included home lockdown, limited freedom of movement in public spaces and between territories, closure of establishments, limitation of commercial and cultural activity, and prohibition of visits to nursing homes [3,4]. 

Aging is a major risk factor for serious illness and death from COVID-19 [5]. O’Driscoll et al. [6] described how Europe had a high incidence of deaths in those over 65 years of age, especially in nursing homes. Therefore, COVID-19 became a geriatric emergency [7]. As a result, exceptionally severe social isolation measures were applied in nursing homes, and in activities considered risky for the elderly population. For example, going outdoors was often limited and therapies were suspended. Residents experienced longer stays in their rooms [8,9,10], causing a significant occupational imbalance. In Germany, Schweighart et al. [10] described how the absence of activities during social isolation caused discomfort, restlessness, and mental impairment [11]. In Spain, Becerra-García et al. [9] described how in nursing homes, resuming activities after the pandemic improved the residents’ depressive symptoms. 

Attempts have been made to protect institutionalized persons through restrictive measures such as social isolation. Isolation and loneliness are risk factors for mental and physical health [12]. Throughout the pandemic, residences used both technology (such as tablets, smartphones, etc.) and conventional therapies (such as laughter therapy, horticultural therapy, and reminiscence therapy) to maintain the occupational balance of the elderly, arouse the feeling of usefulness, and generate positive feelings during the period of social isolation [13].

The aim of the present work was to analyze the lived experience of elderly people in residence during lockdown and social contact restrictions resulting from the COVID-19 pandemic.

## 2. Materials and Methods

For this qualitative study, two standards were followed: Standards for Reporting Qualitative Research (SRQR) [14] and Consolidated Criteria for Reporting Qualitative Research (COREQ) (https://www.equator-network.org/ accessed on 1 October 2022) [15]. 

### 2.1. Study Design

An exploratory qualitative study was conducted based on Sandelowski’s proposal [16,17,18,19]. The theoretical framework that guided this study was interpretivist [20]. The aim of this study was to understand the beliefs, values, and motivations underlying individual health behaviors [21]. 

Exploratory qualitative studies can be identified by three main features. First, such studies aim to identify a critical event or situation. They then seek to show ‘what is happening’ and ‘how it is happening’ through qualitative methodology [18,19]. Lastly, exploratory qualitative studies aim to provide a comprehensive summary of events in everyday terms which resemble the qualitative data collected. This design is the method of choice when direct descriptions of phenomena are desired, such as in the present study [16,17,19]. 

This study was approved by the Ethics Committee of the Rey Juan Carlos (code 0105202011020) and had the permission of the center where the study was conducted.

The research team consisted of 4 occupational therapists, 1 nurse, and 1 physical therapist. Five of the researchers had professional experience in the field of geriatrics. Four of them had professional experience in the development of qualitative studies in health sciences.

### 2.2. Participants, Sampling Strategies and Sample

Participants were recruited from a geriatric residence in the province of Cáceres. A purposive sampling [22] was carried out to find residents who could provide more information in relation to the research question.

Inclusion criteria were (a) to be older than 65 years of age; (b) to have been institutionalized in residence during lockdown (February–November, 2020); (c) to be without cognitive impairment (Mini Mental State Examination ≥ 27 points) [23]; and (d) to have signed the informed consent.

Exclusion criteria: (a) People with dementia or cognitive impairment; (b) People with communication deficits; (c) People who do not maintain all their legal rights and capacities for autonomy and/or who are represented by a legal guardian; (d) People who refuse to participate in the study and/or refuse to sign the informed consent. Of 75 residents, 41 were interested in participating, and 29 met the criteria. Twenty-four participants were included in the present study, and data collection was terminated when redundancy of information was reached [24]. Previous qualitative studies [24,25] described how the total number of participants included does not depend on a previous calculation of the sample size, rather it is based on the saturation or redundancy of the information obtained in the interviews. Turner-Bowker et al. [26], covering the estimation of sample size in qualitative studies, pointed out that 97.2 and 99.3% of the contents emerge between interviews 20 and 25.

### 2.3. Data Collection

In-depth interviews (unstructured and semi-structured interviews) and the researcher’s field notes were used as data collection instruments [27]. Data collection was conducted in two phases between May 2020 and December 2021. The first phase of data collection was conducted using unstructured interviews (participants 1–8) between May and June 2020. This type of interview is the most appropriate when there is not a great deal of knowledge of the phenomenon [22]. The participants were asked, “What was the experience of residential lockdown like for you during the health crisis caused by COVID-19?”.

The second phase, developed between February and December 2021, was conducted through semi-structured interviews (participants 9–24) based on the analysis of participants’ responses from the first phase. The question guide for the semi-structured interviews is shown in Table 1.

The interviews were conducted in person by A.C.-M., and subsequently transcribed for analysis [27]. A total of 24 interviews were conducted (one per participant). The average interview duration was 45 min, and the total interview duration was 1092 min. In addition, 24 field notes were collected during the interviews. These provided a rich source of information, as participants were able to describe their personal experiences beyond those requested in the semi-structured interviews. The field notes also allowed us to reflect on the different methods of data collection utilized in this study.

### 2.4. Data Analysis

An inductive thematic analysis was performed [28]. During coding and reviewing the transcripts, the relevant narrative segments of the interview were identified and assigned codes (meaningful units). They were then grouped into clusters with common meanings (categories), until the themes and subthemes were identified by consensus among the researchers [22]. The analysis of the interviews was complemented with the researcher’s field notes. No qualitative data analysis software was used. The interviews were double-coded by two researchers (A.C.-M. and J.P.-C.).

### 2.5. Rigor

Lincoln and Guba’s four criteria for trustworthiness [29,30] were followed: Credibility; Transferability; Reliability; and Confirmability. In addition, the following five techniques were applied to enhance rigor: (a) triangulation of data collection instruments; (b) triangulation of researchers during analysis (double-coding); (c) member checking with participants after the interviews, with no additional comments from any of the participants; (d) detailed description of the research study, researchers, and participants; and (e) records of researchers’ reflexivity during the study.

## 3. Results

Twenty-four participants (18 females and 6 males) were included, with a mean duration of stay of 36.29 months when lockdown began. The mean age was 81.29 years (S.D. 9.005). The remaining clinical and sociodemographic data are in Table 2.

The thematic analysis identified two main themes: (1) Emotional impact of the experience of COVID-19 lockdown (subthemes: Contradictory feelings; Illness and death; the Importance of routine; Feeling busy; and the Role of religious beliefs); and (2) Support as a therapeutic tool (subthemes: Family support; Peer support; and Support from professionals). The classification of themes and subthemes is shown in Table 3.

### 3.1. Theme 1: Emotional Impact of the Experience of Lockdown by COVID-19

Participants described experiencing contradictory feelings during the pandemic, particularly in relation to illness and death, and to changes in routine. Residents emphasized the importance of feeling busy and the role of religious beliefs while they coped with lockdown.

#### 3.1.1. Subtheme: Contradictory Feelings

The residents described feeling sad, overwhelmed, and anxious during lockdown. Interviewees described the sadness and grief associated with the separation from loved ones, as well as a fear of dying without the chance to say goodbye to family. The participants described how these worries made it difficult for them to sleep:

”*It makes me very sad, because I am looking forward to seeing my children*”;(P.3)

”*There have been nights when I haven’t slept…just thinking about this*”.(P.7)

Some residents chose to hide these feelings from their peers, family members, and nursing home professionals:

‘*They tell me*‘*you have a lot of joy*,’* but deep down I have sadness*’;(P.1)

‘*I haven’t told anyone, I didn’t want to say anything*’.(P.13)

At the same time, participants felt joy to talk to people and described a sense of calm in knowing that virus containment measures were working. They described the peace of mind in knowing that there were little to no cases of COVID-19 within the residence, and that those that did arise were mild:

*‘I try to talk a lot with people... that gives me joy and I overcome the sorrow I may have inside me’*;(P.1)

*‘I felt calm because they treated us very well... that’s how it was, because here it* [the virus] *entered and nothing happened despite everything that could have happened’*.(P.20)

Other residents were not afraid of the disease, nor of death. From their perspective, it was necessary to be careful, cautious and respectful of the virus, and they had to live with this situation and deal with it in the best possible way.

‘*I have not been afraid, I have simply been cautious*’;(P.4)

‘*I’m not afraid of the virus, no, I have to keep fighting*’.(P.10)

#### 3.1.2. Subtheme: Illness and Death

Residents thought about the possibility of becoming sick or even dying. Some felt prepared, while others described fear and worry. Some of the participants explained how they left it in God’s hands when their time would come, while also maintaining their drive to live:

”*Life can go in a moment, and you have to be prepared*”;(P.1)

”*I do not want to die because I love life very much…I am not prepared *[to die]*, I was very afraid* [of the virus]”;(P.13)

“*…we have nothing left to say but let it be what God wants and let him have us when he wants us…I still want to live very much*”.(P.19)

”*You cannot be afraid of death because it is something we cannot control, you can be careful, but not afraid, you have to face things*”.(P.16)

In relation to the possibility that one of their family members might die, the residents said that they would prefer to die before one of their family members, especially among those participants with children and grandchildren:

“*I was praying to God that I would be the first to die before my children. I couldn’t stand it *[she gets emotional]. *Even for my grandchildren I would give my life before anything happens to them*”.(P.23)

It should be noted that the residents described how the meaning of the disease evolved throughout the pandemic. At first, in case of contagion, it was always certain death. Later, they found that it was survivable, and in many cases, it was mild or even asymptomatic:

”… [the virus entered the residence]* and the first days were horrible because of the fear, but then it was not so serious. There were only a few sick people here and as the days went by, I relaxed… we were not all going to die as I thought*”.(P.24)

#### 3.1.3. Subtheme: The Importance of Routine

Residents recounted how their occupational routines were altered and changed by the pandemic, and the resultant feelings of frustration. Participants mentioned that changes in their routine included the addition of more scheduled activities such as cognitive training, psychomotor programs, and leisure programs. Above all, participants noted the loss of interaction with other residents, such as sharing meals. To this they added the loss of freedom of movement due to lockdown in the rooms, and the prohibition of visits from their families. They all wished to regain their previous routines, their pre-pandemic “normalcy”, which they associated with happiness and joy. However, they said they had been able to accept the new situation brought about by the pandemic:

“*Routine makes you happy and you want to get it back*”;(P.1)

“*Without my daily activities I was broken, I missed it a lot. The pandemic changes many things*”;(P. 10)

“*I thought I was dying in there and at first, I thought it was going to be 2–3 days, but I watched the days go by with resignation and I got used to it, but it was horrible. For me, the freedom to move and relate to others is the most valuable thing a human being has and that was taken away from us*”.(P.24)

#### 3.1.4. Subtheme: Feeling Busy

Residents recounted the importance of occupational performance during the pandemic. Even though activities were suspended, feeling busy served to distract them from thinking about the pandemic. Being busy made them feel that time passed more quickly, and they found that fewer negative thoughts arose:

“*I had the need to be doing things, when one was better occupied, the thoughts were softer. Doing nothing can make people crazy*”.(P.17)

All residents were aware of the importance of keeping occupied. However, some described that they were not able to continue doing certain activities, due to a lack of concentration, motivation, or energy:

“*I didn’t do them* [occupations] *because I didn’t feel like it, I wasn’t able to focus and there was no way to get me out of everything I was thinking. I think it would have been better for my head, but the spirit was lost*”.(P.23)

#### 3.1.5. Subtheme: The Role of Religious Beliefs

The participants described religious beliefs as an important factor in coping with the experience of lockdown emotionally. Many of the residents noted that they feel grateful to God for not being sick, that there were no contagions in the residence, and that their family members were in good health:

“*We have to thank God because our families are fine, I myself am fine. Thank God that we have not had any problems in the residence*”;(P.2)

“*Blessed be God, nothing has happened to us*”.(P.3)

In addition, residents reported that when they prayed, they saw God by their side. They felt comfortable and calm, and they felt heard when they were alone. Further, participants described how praying helped them to let off steam:

“*Beliefs are very important because you feel listened to and even more so when you are alone*”;(P.9)

“*Every night of this pandemic, but every single night,* [my faith in God] *made me feel satisfied and comfortable*”.(P.10)

### 3.2. Theme 2: Support as a Therapeutic Tool

Residents emphasized the importance of the support they received from family members, fellow residents, and professionals. For them it was an essential treatment tool received in different formats.

#### 3.2.1. Subtheme: Family Support

Phone calls from their relatives provided support to the residents, and helped them overcome sadness. They described how having someone listen lifted a sense of burden, that through conversation they reclaimed the will to live. Family support brought back a sense of happiness they had lost, and the will to keep fighting. Residents explained that talking to their family reassured them, made them feel supported, and increased their confidence and sense of strength to face lockdown:

“*That is very sad, because to get over it a little I call my husband when I come home from breakfast, when I come home from lunch and at night before I go to bed, I call him*”;(P.1)

“*They support you and tell you no mom, don’t worry because this* [the lockdown] *is like that. You already feel a relief with that support and that confidence. That’s what helped you to live, to feel listened to*”;(P.3)

”*That really made me very happy, I let off steam with them*”.(P.14)

Some of the participants described that while talking to their families they felt cheerful and encouraged. However, the end of the conversation was bittersweet: they felt uneasy, sad, and helpless:

“*That time I was crying like Magdalena* [Spanish expression that means to cry inconsolably] *when I hung up with them, I felt like the worst of impotence, they tried to cheer me up, but I was very upset*”;(P.19)

“*Notice how rare it was that they would call me, and it gave me a lot of joy, but when I hung up* [the phone] *I would get very sad and upset about being alone again. It was a bittersweet taste, I couldn’t wait for them to come, but when they left, I felt bad*”.(P.22)

However, there were participants who felt the lack of support even more, in cases where their loved ones had passed away or where family members did not care about them:

”*My sister’s support has always been missing from me and now even more so. I would have loved for my mother to have been there, she would have been my support, I have missed her much more*”.(P.9)

”*They have barely called me on the phone, that is sad that your family does not bother to call you daily or anything*”.(P.21)

#### 3.2.2. Subtheme: Support from Fellow Residents

Residents reported how the support received from other residents helped them to grow in the worst moments and to feel listened to. This support made them feel comfortable and made it easier to establish good relationships. The roommates in the residence were, on many occasions, the only contact they had, and served as a pillar of stability during the pandemic: 

“*My roommate’s support has been great and very important. We were both alone in the room*”;(P.14)

“*The friendship I have had with my roommate has been very good for me because if it hadn’t been for her friendship, I would have died here of disgust… surrounding myself with her has taught me many things, the truth has been a very big support in this time because thanks to her I’ve gotten over her better* [the lockdown]”;(P.21)

“*We told each other how we felt the stresses, the sorrows, and everything. If I had gone through it alone, I would have had a much worse time*”.(P.22)

However, there were participants who did not feel supported by other residents. They described wishing they had had a companion to lean on more, share concerns, and avoid feeling alone. They explained how, with more support, the pandemic would have been easier to endure:

“*It would have been nice to have been with someone with whom I could have talked and with whom we could have supported one another, I missed that a lot*”;(P.12)

“*Another roommate would have been better, with my roommate it was very difficult because I could not tell her anything serious, she was like a wall, it was like being alone*”.(P.23)

#### 3.2.3. Subtheme: Support from Professionals

The residents appreciated the support given by the nursing home staff. They described how the professionals tried to create relaxed moments even though they were tired from work. The resident staff listened to them, stayed close by, and were always willing to help them. In addition, the residents described how the nursing home staff supplied the support that their family members could not give them, and some of the participants even reported that it was as if the professionals were their own family:

“*They* [the nursing home staff] *came many times and listened to me and it suited me better*”;(P.10)

“*I was very grateful for this support because my family could not be there, so they* [the nursing home staff] *were there*”;(P.12)

“*During these days when they* [the nursing home staff] *were there, they were my brothers who sheltered me and took care of me, they listened to me. Having them as a family was very important, I felt good, and I felt understood*”.(P.16)

There were also participants who described that, after contact with the staff, they returned to their previous state of discomfort. Others described how they wished the staff would have spent more time with them, sitting and talking quietly, without schedules: 

“*Those moments with* [the nursing home staff] *were like a breath of fresh air, but then they would leave, and I would get really sick again*”;(P.19)

“*They would come and listen to us because they can’t say otherwise, but I have missed them sitting peacefully and talking to us because we needed that half-hour of patient presence and conversation*”.(P.17)

The participants reported that they must be grateful for the great work carried out by the professionals despite their own emotional state. The participants understood that it was often difficult for the staff to care for them. This empathy for the professionals helped the residents feel better themselves:

“*We understand the effort and work that they* [the nursing home staff] *do to keep us better and to heal us. It is very difficult and we must thank them for that after everything they have done*”;(P.10)

“*I saw one of them* [the nursing home staff] *upset and I worried about them too, because I saw that they were very nervous. Even so, they always came with a smile to cheer us up, and for that we must thank them*”.(P.14)

## 4. Discussion

The participants in our study described experiencing conflicting feelings during the social restrictions caused by the pandemic, placing high importance on occupations, disruption of routines, and social support.

Our results show how residents exhibited contradictory feelings about illness, death, and occupation. Participants described feeling sad, feeling overwhelmed, and fearing death. Previous studies [31,32,33,34] have shown how the pandemic led to frequent feelings of sadness, anxiety, depression, and restlessness. These studies also found that this continuous worry led to sleep disturbances [35]. 

The residents reported that despite the many negative experiences during lockdown, they were happy that they could continue to relate to each other. They were also happy to discover that the appearance of cases with COVID-19 did not mean death. Our results coincide with the study by Van der Roest et al. [31], where people institutionalized during the pandemic also perceived moments of tranquility. 

On the other hand, our results reveal that some participants concealed their true feelings from their caregivers and/or relatives. This could be due to the need for self-protection, or to protect those around them from suffering [36]. Idoiaga et al. [37] suggested that this could be due to an emotional “shock”, resulting in the inability to manage and express these feelings.

Our results describe how residents frequently thought about the possibility of becoming ill or dying. These findings are consistent with previous studies [35,38], which described how, during isolation, COVID-19 provoked recurrent fears, distress, and thoughts about not having a dignified death. However, in our results there were residents who remained cautious given the seriousness of the situation but had not experienced fear. Our results show a sense of demotivation and apathy during social lockdown, as well as a sense of nostalgia for lost routines. Previous studies [8,39,40] have shown how routine can generate feelings of happiness and well-being and can also serve as a distraction from problems [8,39]. Hou et al. [40] identified that the disruption of these routines can lead to distress and sadness. Kaelen et al. [41] found that residents, when faced with this disruption of routines, felt a deprivation of freedom, as if they were in prison, accompanied by feelings of claustrophobia and depression. This coincides with our results, where residents described how they felt overwhelmed, and felt a lack of freedom.

The studies by Gismero-Gonzalez et al. [42] and Boekel et al. [43] describe how the return to a routine similar to the previous one is easy and quick. Malkawi et al. [44] emphasized that, during the pandemic, people increased the number of meaningful activities and daily chores, because there was a positive relationship between the performance of meaningful activities and the mental health of older adults [45], and even a sense of protection from COVID-19 [46,47]. In their study on the impact of COVID-19 measures on the well-being of older long-term care facility residents in the Netherlands, Van der Roest et al. found that 53% of participants suffered more apathy during the pandemic [31]. Prior studies frequently recommended maintaining routines and meaningful activities during COVID-19, given the proven benefits to mental health [11,46].

According to our data, residents attached great importance to the religious role during the pandemic. This helped them to feel calm, confident, and accompanied. Religiosity and spirituality among residents is a theme that has already been addressed in other studies [48,49,50,51], which emphasized the need to give importance to the spiritual care of residents. These studies emphasized not only fostering the spirituality of the residents, but also of the nurses in order to better meet this need in the residents. These studies recommend that nursing home managers should promote the spiritual care education of nurses, and work to improve perceptions of spiritual care and spiritual health, to improve their spiritual care competence and to satisfy the spiritual care needs of patients [48]. 

To date, we have not yet found studies that investigated the importance of spirituality in nursing home residents during the pandemic, despite it being such an important part of the residents’ lives. However, we have found such topics among studies with disadvantaged communities [49,52]. For example, Aja et al. [49] showed how the design and implementation of an anti-COVID-19 song of hope within an impoverished community helped individuals to cope with the physical, sociopolitical, psychological, and spiritual trauma caused by the pandemic. Further, the study by Stizzi et al. [52] indicated how pastoral counselling can be useful in situations of stress in highly deprived areas.

Our results show how social support was a key component of the residents´ experiences. The support received encouraged them to overcome sadness, and gave them the will to continue living. Our results coincide with previous studies [13,53,54] which showed that the more frequent the communication, the greater the satisfaction of both family and residents when communicating by telephone or video call. Social support served as a protective factor [55] in that maintenance of these relationships decreased the negative effects of isolation on mental health [40,42].

Our results show how the support received among residents helped them to feel listened to and protected. Zhao et al. [56] pointed out that the greater the contact between residents, the lower the depressive symptoms. Furthermore, our residents highlight the support received by professionals. These results coincide with the study by Lood et al. [8], which described how the healthcare staff provided support and security, and established close contact with the residents during social isolation. 

A study by López et al. [57] described how specific social supports, such as positive family dynamics and strong marriages, are related to psychological well-being. This study emphasized that personal strategies and resilience were key to healthy coping with lockdown.

In comparison, other studies demonstrated other types of coping mechanisms employed by older adults. These included emotional eating [58], increased alcohol consumption, increased smoking [59], and utilizing open spaces at home [60,61].

Additionally, according to our data, residents were grateful. Gratitude can be of great benefit. In their study on gratitude, Büssing et al. [62] pointed out that gratitude is trainable and can facilitate an emphasis on the positive aspects of life despite difficult times, which in turn aids overall well-being.

The present study had limitations. Firstly, the results obtained cannot be extrapolated to the entire institutionalized population due to the nature of the qualitative methodology [20]. Secondly, the present study included 24 participants. The sample size in qualitative research does not depend on a previous calculation but was established through redundancy of information or saturation during the collection of data from the interviews [24,25]. In addition, Turner-Bowker et al. [26] reported that 97.2–99.3% of the saturation can be analyzed between interviews 20 and 25. Nevertheless, they may provide relevant information to assess the impact of the pandemic on residents’ occupations and feelings.

## 5. Conclusions

In conclusion, our results highlight the importance of occupation and routines during isolation. In addition, the support received from other residents, from professionals, and from their loved ones via telephone was an essential pillar for moving forward during the pandemic. These results may help in understanding the experiences of institutionalized persons during the major health crisis caused by COVID-19, to describe their feelings and the importance of occupation, routine, and social support. Future studies need to delve deeper into the experiences of older people in other contexts and their effect on occupation during the COVID-19 pandemic.

## Figures and Tables

**Table 1 ijerph-19-15712-t001:** Semi-structured interview question guide.

Research Areas	Questions
Emotional impact of the experience	Thinking back to the beginning of the pandemic and up until now, has the pandemic impacted you emotionally? If so, how? How did the possibility of people in the residence becoming infected during lockdown make you feel? Have you been affected in any way by being considered an at-risk population? If so, how?
Importance of routine and occupation	Did the pandemic impact your daily routines? If so, how did this change in routine affect you? How would you describe your time during the pandemic? Did you feel busy? Unoccupied? Please elaborate. What activities did you engage in during the pandemic? What did they mean to you? During lockdown, did you miss participating in any activities in which you were previously involved? If so, which ones? If so, how did this make you feel? Have you given up any activities during the pandemic? Which ones? How did this impact your experience of lockdown?
Support received	Have you had any social/family/professional support during the pandemic? How has that support made you feel and how has it impacted your experience of lockdown?

**Table 2 ijerph-19-15712-t002:** Summary of clinical and sociodemographic data.

Code	Sex	Age	Marital Status	Diagnosis	Mini-Mental Score	Months Admitted 01/03
P1	M	89	Married	Parkinson’s	30	44
P2	M	87	Widow	Obesity, poor return circulation	29	30
P3	M	83	Widow	Diabetes, heart disease, and arthritis	30	47
P4	M	79	Widow	Diabetes, maculopathies	28	35
P5	H	82	Single	Malignant neoplasm of prosthesis, hearing loss	29	45
P6	H	67	Single	HIV, COPD	28	42
P7	M	92	Widow	No diagnosis	30	30
P8	H	69	Divorced	COPD, ex-alcoholism	29	30
P9	M	68	Divorced	OCD, ex-alcoholism	28	26
P10	H	68	Single	Left carotid lacunar carotid ischemic stroke	30	31
P11	M	77	Single	Hepatitis C	29	46
P12	M	69	Single	Multiple sclerosis	28	40
P13	M	90	Widow	Depression, HTA	28	36
P14	M	92	Widow	Breast cancer, HTA	28	32
P15	M	94	Widow	Dyslipidemia and vertigo	29	39
P16	H	78	Widow	Hypertension, dyslipidemia, obesity, cataracts.	28	38
P17	M	93	Married	Hypoacusia, degenerative maculopathy	28	37
P18	M	85	Widow	Dyslipidemias, bronchial asthma	28	30
P19	M	87	Widow	Ischemic stroke, mitral insufficiency, chronic anemia	30	36
P20	H	83	Married	Thalassemia, bilateral gonalgia	29	27
P21	M	68	Single	Renal disease, HTA	28	44
P22	M	88	Married	Diabetes mellitus II, osteoarthritis of the knee	29	31
P23	M	82	Widow	COPD, dyslipidemia, diabetes mellitus II, dysphagia	28	45
P24	M	81	Widow	Hip fracture, vertiginous syndrome	28	30

Note: HIV = Human Immunodeficiency Virus; COPD = Chronic Obstructive Pulmonary Disease; HTA = Health Technology Assessment.

**Table 3 ijerph-19-15712-t003:** Main themes and subthemes.

Topics	Subtopics
1. Emotional impact of the experience of lockdown by COVID-19	1.1. Contradictory feelings
	1.2. Illness and death
	1.3. The importance of routine
	1.4. Feeling busy
	1.5. The role of religious beliefs
2. Support as a therapeutic tool	2.1. Family support
	2.2. Support from peers
	2.3. Support from professionals

## Data Availability

The personal data related to this study are stored in the data protection file belonging to the Rey Juan Carlos University. Considering the qualitative nature of this study, we are unable to provide the transcribed files, in compliance with the Spanish Protection of Personal Data and Information Act (1999).

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
