# Peer review of "Emotional Impact and Perception of Support in Nursing Home Residents during the COVID-19 Lockdown: A Qualitative Study"

_ijerph, 2022, doi:10.3390/ijerph192315712_

Round 1

Reviewer 1 Report

Thank you for the opportunity of reviewing this manuscript about the emotional impact and perception of support in nursing home residents during the COVID-19 lockdown.

Overall, the article is written clearly, and it is appropriately divided into sub-chapters. However, language-wise, the article requires some editing and proof-reading. The manuscript needs to be extensively edited for grammar and spelling For example, deaths line42, researcher's line 96, were followed line 122, relaxed line 260…

The abstract needs revisions to better present the content of the study, including its important findings.

Methods : Please namet he province, Why XXX?

Exclusion criteria shouldn't be everything they aren't inclusion criteria.

In qualitative analysis, 'categories' are used instead of 'themes'.

Author Response

We would like to thank the editor and reviewers for their comments in this review, which have greatly improved the readability of the manuscript. We would like to inform you that we have edited the manuscript according to the very constructive suggestions from the reviewers.

Below, please find a list of revisions and a response to each of the reviewer’s comments. We hope that the revisions in the manuscript and our accompanying responses will be enough to make our manuscript suitable for publication in the IJERPH.

We shall look forward to hearing from you at your earliest convenience.

Yours sincerely,

The authors

Reviewer 2 Report

I had the pleasure to read the paper entitled "Emotional impact and perception of support in nursing home residents during the COVID-19 lockdown: a qualitative study", which analyzed the lived experience of 24 elderly nursing home residents during the confinement and social contact restrictions resulting from the COVID-19 pandemic. The topic was very interesting and informative. But I had several concerns about this manuscript.

1. This study only included 24 participants in nursing homes. The empirical study did not give sufficient information and discussions related to the research topic. Therefore, the number of respondents should be appropriately increased and the sampling problem should be discussed in more detail.

2. Line 142-219, this study provided with evidence for the emotional impacts of the experience of COVID-19. All these results are consistent with the existing research. More specific analyses are needed to reveal some different impacts from other studies.

3. Lines 223-275, this study identified three types of support for coping with the emotional impact of the experience of COVID-19. However, the authors should provide with more details on the dynamics of the supports reducing the negative emotions and feelings. It is recommended to include more interviews here.

4. Lines 277-332, the part in the article needed more discussions. The supports have been also fully discussed in the existing research. The authors should tell us more about the different findings from the normal social supports showed in other studies.

Author Response

(The authors gave the same response as above.)
